# Solid-to-liquid phase transitions of sub-nanometer clusters enhance chemical transformation

Juan-Juan Sun[1] & Jun Cheng [1]*

Understanding the nature of active sites is crucial in heterogeneous catalysis, and dynamic changes of catalyst structures during reaction turnover have brought into focus the dynamic nature of active sites. However, much less is known on how the structural dynamics couples with elementary reactions. Here we report an anomalous decrease in reaction free energies and barriers on dynamical sub-nanometer Au clusters. We calculate temperature dependence of free energies using ab initio molecular dynamics, and find significant entropic effects due to solid-to-liquid phase transitions of the Au clusters induced by adsorption of different states along the reaction coordinate. This finding demonstrates that catalyst dynamics can play an important role in catalyst activity.

[1] Collaborative Innovation Center of Chemistry for Energy Materials (iChEM), State Key Laboratory of Physical Chemistry of Solid Surfaces, College of Chemistry and Chemical Engineering, Xiamen University, Xiamen 361005, China. *email: chengjun@xmu.edu.cn

The concept of active site in heterogeneous catalysis was coined by Taylor[1] nearly a century ago, and ever since chemists have made great endeavors to characterize the structures of the active sites of catalysts and elucidate how the special arrangements of atoms accelerate chemical reactions[2,3]. The rationale behind is that molecular understanding of the active sites would help design and search for better catalysts[4]. A prominent example is the Au catalyst[5,6]; it had been long thought of as an inert material incapable of catalyzing any reactions[7] until Haruta et al. discovered that it becomes active when being made into nanoparticles[6,8]. It is now well established that the stepped or kink sites abundant in nanoparticles consist of under-coordinated Au atoms that are active in breaking chemical bonds.

The notion of sites however often gives a somewhat static or rigid picture of the active sites of catalysts with fixed atomic configurations. Although in recent years the advent of in situ spectroscopic and microscopic techniques[9–13] and electronic structure calculation methods has allowed for investigating the dynamic evolution of the structures of catalysts under reaction conditions[14–19], the static perspective has been still taken in identifying the active sites under environmental conditions and monitoring the transformation of one type of active sites into another. For example, ab initio molecular dynamics (AIMD) simulations have been used to explore the diversity of structural configurations of clusters, and however elementary reactions have been still calculated on given cluster structures using static optimization techniques at 0 K[15]. This is justifiable because from the viewpoint of elementary reactions, which occur at the time scale of picoseconds, the catalytic sites are essentially static considering the catalyst structures usually evolve at macroscopic timescales[10,12,20]. The question arises, what if the time scale of the dynamic evolution of catalyst structures overlaps with that of chemical reactions? In the following, we will show that this is indeed the case for the reactions occurring on sub-nanometer clusters, and moreover we find an anomalous decrease in the reaction free energies and barriers owing to entropic effects that are attributable to (quasi-)first-order solid-to-liquid phase transitions of catalyst clusters during the course of the chemical reactions. This discovery provides a new insight into our understanding of catalysis on small clusters widely existing in supported catalysts.

## Results

**Free energies of $O_2$ dissociation on $Au_{13}$.** Here, we choose $O_2$ activation on small Au clusters as example, because it is a key step in the first reaction (i.e. CO oxidation) demonstrated for the Au catalysis by Haruta et al.[8] and numerous other oxidation reactions[21] such as alkene oxidation[22,23]. We employ AIMD simulation[24,25] which is very suitable for the purpose of both accounting for the chemisorption of molecules on clusters and the dynamic evolution of clusters when breaking chemical bonds. We first calculate the free energy profiles of $O_2$ dissociation reaction on $Au_{13}$ cluster (see Fig. 1a, Supplementary Fig. 1a). The free energy profiles are obtained by computing the potential of mean force (PMF) along the reaction coordinate of O–O distance. It is shown that the temperature plays an important role in the structure dynamics of clusters[26–28], and therefore we investigate the dynamic catalysis on the cluster at a temperature range varying from 120 K to 600 K. We also calculate the reaction at 0 K using static geometry optimization for comparison. As shown in Supplementary Fig. 2, the cluster structure undergoes significant change along the reaction path, more so at higher temperatures, in contrast that it hardly changes at 0 K.

The calculated free energy profiles and the temperature dependence of reaction free energies and barriers are shown in

Fig. 1b, c, and the corresponding PMF data is given in Supplementary Fig. 3. The hysteresis (Supplementary Fig. 3d) is very small in thermodynamic integration and the statistical errors in the PMF (Supplementary Fig. 3e, f) are negligible. It is evident that with the temperature increasing, both the reaction free energy and barrier decrease dramatically (Fig. 1c, Supplementary Table 1 for detailed values). In contrast, such temperature dependence of free energies is not observed by the commonly used geometry optimization method (the hollow circle of Fig. 1c, Supplementary Table 2 for detailed values). The static calculation corrects for entropic terms based on models such as harmonic oscillator, which are largely canceled in energy differences, and hence misses configurational entropies owing to dynamic fluctuation of the cluster structure. In the PMF calculation, AIMD samples the ensemble of the relevant configurations of the cluster consisting of both the adsorbed $O_2$ and the $Au_{13}$ cluster, and therefore takes all entropic contributions into consideration. The effect of structural dynamics is also manifested by the finding that the O–O distance at the transition state (TS) of the reaction free energy pathway decreases from 2.2 to 1.8 Å when the temperature increases from 120 to 600 K, in comparison to the value of 2.1 Å obtained from the static calculation (Supplementary Fig. 4).

It is interesting to note from Fig. 1c that the curves of the free energy and barrier against the temperature show three characteristic regions, with a steep transition region at the temperature range of about 300–400 K separating the low and high temperature regions with small slopes. This behavior is even clearer if taking the derivatives of the free energies with respect to the temperature, which gives the entropy changes as a function of the temperature. As illustrated in Fig. 1d, the entropy changes show Gaussian-like distributions peaked in the temperature range of free energy transition. The peak heights are surprisingly large, on the order of about 1800 J mol$^{-1}$ K$^{-1}$ and 600 J mol$^{-1}$ K$^{-1}$ (i.e., for one mole of the cluster) at ~340 K for the reaction entropy change ($\Delta_r S$) and activation entropy ($\Delta S^{\ddagger}$), respectively. It is these large entropy changes that give rise to the drastic decreases in free energies at the transition temperature range.

**Anomalous reaction entropies and phase transitions of $Au_{13}$.** What causes the temperature dependent behavior of the reaction free energies on the cluster and what is the physical origin behind the enormous entropy changes at the transition region? It is conceivable that such large entropy changes often occur in phase transitions[29]. We thus explore how the total energies change with varying temperature, and calculate the canonical caloric curves $\langle E \rangle (T)$ of the reactant, TS and product, as shown in Fig. 2a–c (also see Supplementary Fig. 5 for the convergence of time averages of total energies). All three caloric curves show a similar trend that the total energies linearly increase with increasing temperature at low and high temperature ranges, with sudden jumps in between as indicated with gray areas in Fig. 2a–c. This is indeed characteristic of phase transitions, and consistent with the common view that small finite-size systems like clusters can show quasi-first-order phase transitions, in which there is no well-defined transition (melting) temperature but rather a range of temperature where both phases coexist[30]. What is intriguing is that the three curves have different transition temperature ranges, which can be clearly revealed from the specific heat curves $Cv(T)$ by taking the derivatives of the caloric curves $\langle E \rangle (T)$ against the temperature. As illustrated in Fig. 2e–g, unlike the bulk limit of a first-order phase transition in which the specific heat is infinite and discontinuous at the melting point, the specific heat curves $Cv(T)$ are continuous and have peaks with finite widths. For convenience, we define the temperature at the peak maximum as

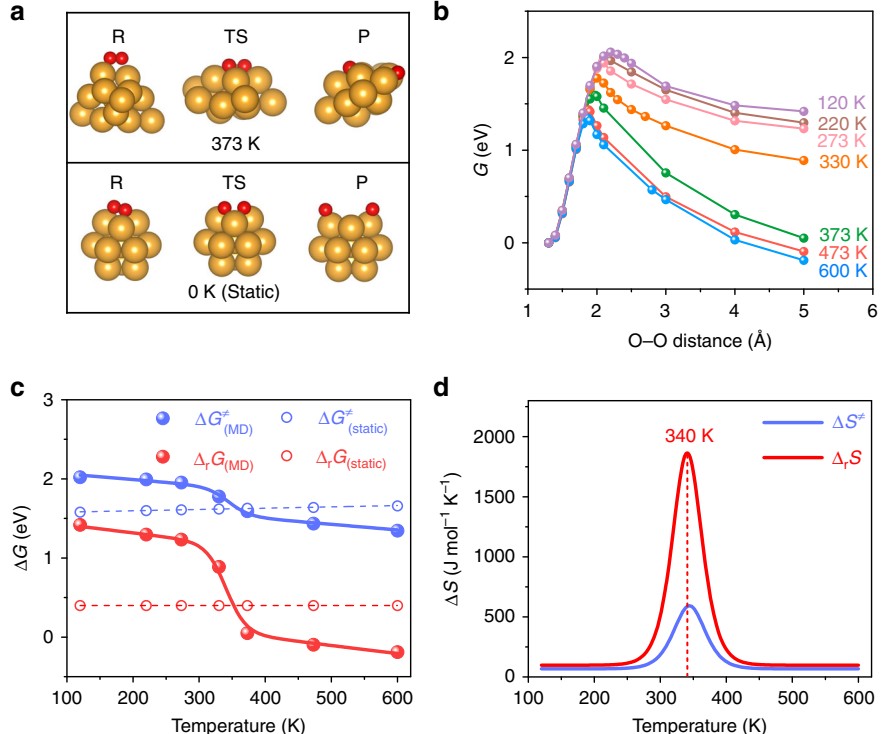

**Fig. 1** Reaction free energies of $O_2$ dissociation on $Au_{13}$. **a** Upper: snapshots of the structures calculated using AIMD at 373 K. Down: structures calculated by static geometry optimization. R, TS and P denote the reactant, transition state and product state, respectively. The balls in yellow and red represent Au and O atoms, respectively. **b** The free energy profiles of $O_2$ dissociation on $Au_{13}$ under different temperature calculated using AIMD. **c** Temperature dependence of reaction free energies ($\Delta_r G$, red) and free energy barriers ($\Delta G^{\ddagger}$, blue). Free energies are referenced to those of the reactant states. The filled and hollow circles represent the calculated energies using AIMD and static geometry optimization method, respectively, and the solid and dash curves are the corresponding fits. **d** Temperature dependence of the reaction entropy changes ($\Delta_r S$) and activation entropies ($\Delta S^{\ddagger}$) calculated by AIMD. The insert value indicates the temperature of maximum entropy change.

the melting temperature in the quasi-first-order phase transition. Thus, the melting temperature for the reactant state is about 390 K, similar to that of bare $Au_{13}$ (~413 K)[31], higher than that of the product state (~340 K), and the melting temperature of TS is in between (i.e., ~350 K).

These differences in the melting temperatures for different states along the reaction coordinate can have great impact on the entropy and free energy changes. Comparing the phase changes of the reactant and product, at the temperature below ~290 K both the reactant and product states are solid, leading to a minor $\Delta_r S$ (the white area in Fig. 2d). When increasing the temperature to the range 290–340 K, the product becomes the solid-liquid coexistence state (the orange area in Fig. 2e), with dramatic entropy increase of the product, while the reactant remains in the solid state (the white area in Fig. 2f). Consequently, $\Delta_r S$ will increase (the orange area in Fig. 2d), to the same extent as the entropy increase in the phase transition of the product. When temperature increases to 340 K, the reactant starts to melt, reaching the solid-liquid coexistence state (the green area in Fig. 2f). The entropy increase of the reactant thus compensates that of the product, leading to reduction in $\Delta_r S$ (the green area in Fig. 2d). When the temperature is higher than 420 K, both the reactant and product completely transform to the liquid states, resulting in a small $\Delta_r S$ (the white area in Fig. 2d). Similarly, the temperature dependence of the activation entropy $\Delta S^{\ddagger}$ (shown in Fig. 2h) can also be explained with the different phase transition behaviors of the reactant (shown in Fig. 2f) and TS (shown in Fig. 2g).

The underlying mechanism of the anomalous entropy change is best illustrated in the Fig. 2i. The small-size cluster is subjected

to the strong influence of the adsorption of reaction species. The change of the state of the adsorbate (e.g., $O_2$ molecule vs two O atoms) could even alter the phase transition temperature of the cluster. Discrepancy between the melting temperatures of the cluster with different states of the species will inevitably result in a transition temperature range in which one state melts while the other doesn't, and hence the anomalous increase in the reaction entropy. Thus, it is possible to search for the favorable temperature range in which the cluster undergoes a solid-to-liquid phase transition along the reaction coordinate, lowering the reaction free energy and barrier.

**$O_2$ dissociation on $Au_8$/MgO and $H_2O$ dissociation on $Au_{13}$.** In heterogeneous catalysis, many active catalysts are metal clusters dispersed on some supports. We therefore further study $O_2$ dissociation on the $Au_8$/MgO model[32] using AIMD. Similar to $Au_{13}$, it is found that the structure of $Au_8$ supported on MgO calculated by AIMD at finite temperatures is rather dynamical, and different from that obtained by static geometry optimization (Fig. 3a and Supplementary Fig. 1c). With the increase of temperature, both the reaction free energy and barrier decrease (Supplementary Fig. 6b, c). In particular, the temperature dependence of $\Delta_r S$ on $Au_8$/MgO (Fig. 3b) is very similar to that on $Au_{13}$ (Fig. 1d), showing a Gaussian-like distribution with a maximum of ~700 J $mol^{-1}$ $K^{-1}$ around 390 K. This Gaussian peak in entropy again results from the different phase transition temperatures of the reactant and product (Fig. 3b–d, Supplementary Figs. 7 and 8) on the supported cluster, suggesting that this anomalous increase in reaction entropy could also occur on real catalysts. For activation entropy $\Delta S^{\ddagger}$ on $Au_8$/MgO, the peak is somewhat absent, the free

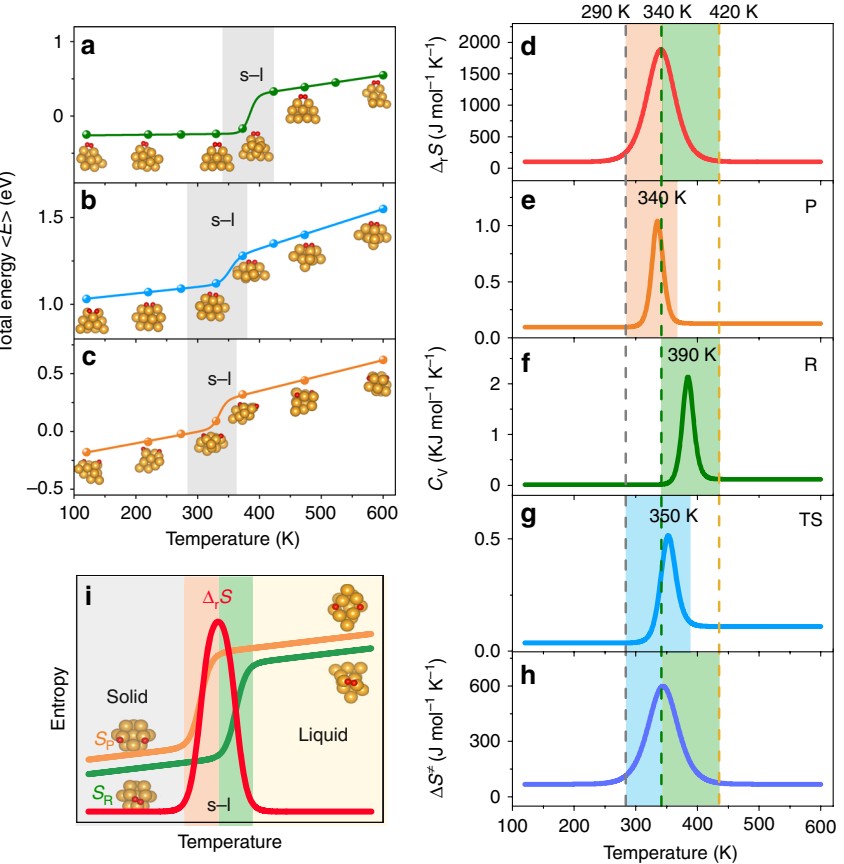

**Fig. 2** Correlation between phase transitions of the clusters and entropy changes. **a–c** Calculated caloric curves ⟨E⟩(T) of the reactant, transition state and product state, respectively. The dots show the calculated values and the lines are fitting curves. The s-l indicates solid-liquid coexistence state. The insets show snapshots of the structures at different temperatures. The balls in yellow and red represent Au and O atoms, respectively. **d** Temperature dependence of the reaction entropy change ($\Delta_r S$). **e–g** The $C_v(T)$ curve of the product (P), reactant (R) and transition state (TS), respectively. The melting temperatures at the peaks are indicated as the inserted values. **h** Temperature dependence of activation entropy ($\Delta S^{\ddagger}$). **i** Schematic illustration of the anomalous reaction entropy change due to the difference in the transition temperature of the solid-to-liquid phase transition of the cluster at the reactant and product state. The curves in red, green and orange indicate the reaction entropy change ($\Delta_r S$), the entropy of the reactant ($S_R$) and the entropy of the product ($S_P$), respectively.

energy barrier decreases almost linearly with temperature (Supplementary Fig. 6c) with a nearly constant activation entropy $\Delta S^{\ddagger}$ of ~60 J mol$^{-1}$ K$^{-1}$ (Fig. 3e). This can be attributable to the similar phase transition temperatures of the reactant and TS, as shown in Fig. 3e–g.

Whether the anomalous entropy change is present (i.e., $\Delta_r S$ vs $\Delta S^{\ddagger}$ on Au$_8$/MgO), or how large is the entropy change (i.e., $\Delta_r S$ on Au$_{13}$ vs Au$_8$/MgO) must depend on the extent of the change of the state of the adsorbed species, and the susceptibility of the cluster to this change. This could explain why the peaks of $\Delta_r S$ are higher than $\Delta S^{\ddagger}$, since the structure of the TS is more similar to that of the reactant than that of the product is. This would also suggest that different types of reactions may show different behaviors. We thus further calculate H$_2$O dissociation reaction on Au$_{13}$ cluster (Fig. 3h). It is found that the general trends of the reaction free energy and barrier (Supplementary Fig. 9b, Fig. 3i) are indeed very similar to those of O$_2$ dissociation (Fig. 1b, c). The entropy changes ($\Delta_r S$ and $\Delta S^{\ddagger}$) of H$_2$O dissociation show gaussian-like distributions but with smaller peak heights than those of O$_2$ dissociation (Fig. 3j and Fig. 1d). The Gaussian peak in entropy results from the different phase transitions of the reactant, TS and product as well (Supplementary Figs. 10 and 11). Comparing the adsorption of H$_2$O and O$_2$ (Supplementary Fig. 1a, b), it is clear that O$_2$ binds to the cluster stronger than H$_2$O, indicating stronger influence on the cluster. Perhaps more

importantly, after dissociation O$_2$ breaks into two O atoms each having two bond valences to bind the cluster, rendering even greater impact on the cluster. While, H$_2$O breaks into OH and H both having single bond valence that can only cast limited impact. Therefore, it is anticipated that not only can the adsorption-induced phase transitions of small clusters improve the activity at certain temperatures but also can shift the selectivity towards certain types of reactions.

## Discussion

This theoretically discovered phase transition effect may have either been present in some experimental work without being realized or not been observed yet since one has to closely look into the interplay between cluster dynamics and temperature. Real catalysts usually have an ensemble of particles with a distribution of sizes, in which small clusters have less isomers[33] and large particles tend to be more rigid[34], both resulting in small entropic effects. Thus, we expect that the dynamic effect will be most pronounced for clusters consisting of a few tens of atoms. Considering that the activity of catalysts results from collective contributions of all clusters with different sizes, there may exist some difficulties in identifying this phase transition effect experimentally. Interestingly, it has been reported that the most active Au clusters for CO oxidation consist of tens of Au atoms[35]. This size-dependent activity of Au clusters may be related to the

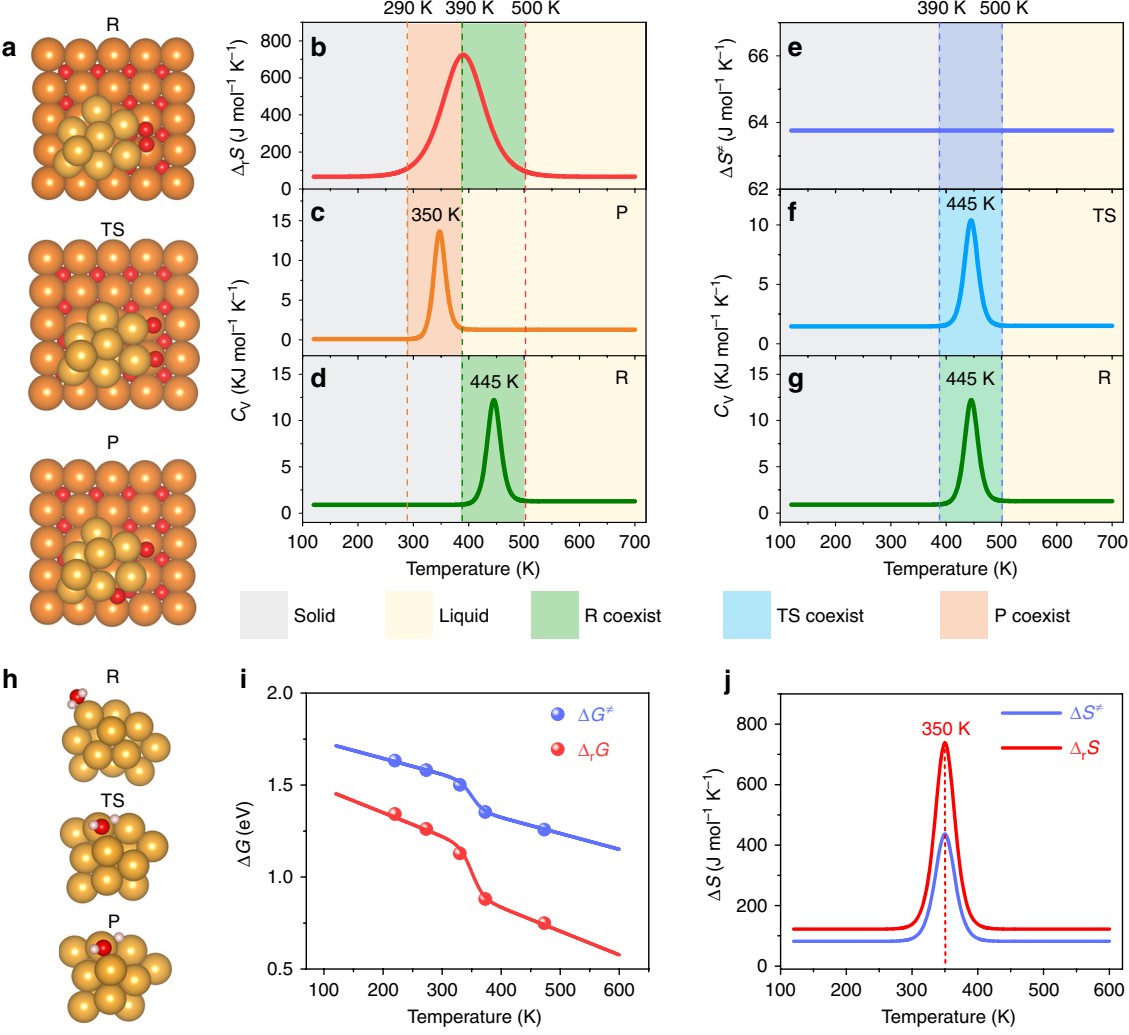

**Fig. 3** Calculation of $O_2$ dissociation on $Au_8$/MgO and $H_2O$ dissociation on $Au_{13}$. **a** Snapshots of the structures of $O_2$ dissociation on $Au_8$/MgO calculated using AIMD at 220 K. R, TS and P denote the reactant, transition state and product state, respectively. The balls in yellow, red and orange represent Au, O and Mg atoms, respectively. **b** Temperature dependence of the reaction entropy change ($\Delta_r S$). **c, d** The $C_v(T)$ curve of the product and reactant, respectively. **e** Temperature dependence of the activation entropy ($\Delta S^{\ddagger}$). **f, g** The $C_v(T)$ curve of the TS and reactant, respectively. The melting temperatures at the peaks are indicated as the inserted values. **h** Snapshots of the structures of the reactant, TS and product of $H_2O$ dissociation on $Au_{13}$ calculated using AIMD at 220 K. The balls in yellow, red and white represent Au, O and H atoms, respectively. **i** Temperature dependence of the reaction free energy ($\Delta_r G$) and free energy barrier($\Delta G^{\ddagger}$). The dots show the calculated values and the lines are fitting curves. **j** Temperature dependence of the reaction entropy change ($\Delta_r S$) and activation entropy ($\Delta S^{\ddagger}$). The insert value indicates the temperature of maximum entropy change.

dynamic effect. Furthermore, supports can significantly influence cluster dynamics. Comparing the results above between $Au_{13}$ and $Au_8$/MgO, it can be seen that MgO can reduce cluster dynamics to some extent and thus the entropic effect induced by phase transition.

However, we envision that with the development of synthesis methods to obtain more uniform sized cluster catalysts and in situ atomic characterization techniques, this phase transition effect could be realized by experiment by investigating the temperature effect on catalyst dynamics and activity. It would also offer possible explanations for experimental observations on the effects of cluster size[36,37], types of supports[38], co-adsorption[39], etc. Understanding these effects on catalyst dynamics will certainly merits future studies.

Finally, we'd like to draw a parallel with enzyme catalysis. First proposed by Linus Pauling[40], enormous acceleration of reaction rates is achieved by enzymes through an enhanced TS stabilization mechanism. This concept has played a key role in drug design for decades, guiding development of catalytic antibodies

that are small, stable molecules mimicking the structures of activated TS complexes in enzymes. However, these antibodies often lead to minor rate enhancements. It is now believed that there was too much focus on rigid TS structures, and protein dynamics (motion) has strong influence on the catalytic activity of enzymes. The latter is becoming the central issue in enzyme catalysis, although its role is still under debate[41]. In analogy with protein dynamics, we show that catalyst dynamics can significantly improve the activity of sub-nanometer clusters through a solid-to-liquid phase transition mechanism, which in turn may shed some light on enzyme catalysis.

To summarize, we study the structural dynamics of small Au clusters and its effects on temperature dependence of free energies of model reactions using AIMD, and discover the abnormal changes of free energies and entropies at certain temperature ranges. We, for the first time, show solid-to-liquid phase transitions of the clusters induced by adsorption of reaction species can facilitate the reactions. Our work highlights the importance of catalyst dynamics in understanding catalytic activity in

heterogeneous catalysis, and opens up promising ways for optimizing the activity and selectivity of catalysts.

## Methods

**Computational models.** The structures of bare $Au_{13}$, and $Au_8$ supported on clean MgO(001) were constructed to study the structural dynamics of the clusters and its effects on catalysis. The $Au_{13}$ cluster was initially built with a highly symmetric cuboctahedral ($O_h$) structure. $O_2$ prefers to adsorb on the hollow site (Supplementary Fig. 1a), and $H_2O$ prefers to adsorb on the top site (Supplementary Fig. 1b). The $Au_{13}$ cluster was simulated in a cubic cell of $15 \times 15 \times 15$ Å$^3$. A four-layer MgO(001)-p($4 \times 4$) slab was used for the support of the $Au_8$ cluster. The cells were modeled under 3D periodic boundary conditions, and the slabs and their images were separated by vacuum with a length of 15 Å. $O_2$ molecule prefers to adsorb at the interface between the cluster and the support (Supplementary Fig. 1c).

**Density functional theory (DFT) calculation.** The AIMD simulations were carried out using the freely available program package CP2K/Quickstep[42] The DFT implementation is based on a hybrid Gaussian plane wave (GPW) scheme, the orbitals are described by an atom centered Gaussian-type basis set, and an auxiliary plane wave basis set is used to re-expand the electron density in the reciprocal space. Perdew-Burke-Ernzerhof (PBE) functional[43] with Grimme's dispersion correction[44] was used. The core electrons were represented by analytic Goedecker-Teter-Hutter (GTH) pseudopotentials[45,46]. For valence electrons ($5d^{10}6s^1$ for Au, $2s^22p^4$ for O, $2s^22p^63s^2$ for Mg, $1s^1$ for H), the Gaussian basis sets were double-$\zeta$ basis functions with one set of polarization functions (DZVP)[47]. We performed the spin-polarized DFT calculations on $Au_{13}$ and $Au_8$/MgO. The spin state of $O_2$ on $Au_{13}$ and $Au_8$/MgO is doublet and singlet, respectively, and the spin state of $H_2O$ on $Au_{13}$ is doublet.

**Free energy calculation.** In the work, we have calculated the reaction free energy profiles by combining AIMD, constrained MD and thermodynamic integration. The reaction free energies (barriers) can be obtained by integrating potentials of mean force (PMF) over a chosen reaction coordinate[48], i.e., the O–O distance of $O_2$ dissociation reaction and the H–O distance for $H_2O$ dissociation. The PMF is calculated using a Lagrange multiplier method, by averaging the force applied on the system to keep the reaction coordinate constant in AIMD runs[49]. In the AIMD simulations, canonical ensemble (NVT) conditions were imposed by a Nose-Hoover thermostat under various temperatures. The MD time step is set to 0.5 fs. In AIMD runs, the trajectories of the first 5–10 ps were regarded as equilibrium periods to ensure equilibria of the systems, followed by another 5–15 ps of production periods for data analysis. For the size of systems studied in this work, the time scale of ~10 ps is sufficient to obtain well converged PMFs. as evident by the time accumulating averages shown in Supplementary Fig. 3a, b. Integrating the forces against the distance gives the free energy profile of the $O_2$ dissociation reaction (Supplementary Fig. 3c). In the force-distance curves, three points crossing the force zero correspond to the reactant, transition and product state, respectively. Note that the hysteresis in our thermodynamic integration is very small. As shown in Supplementary Fig. 3d, when calculating the mean force at the O–O distance of 2 Å, we have started the structure models in both forward and backward directions, i.e., the models with O–O distances of 1.9 Å and 2.1 Å, respectively. The corresponding averages of the mean forces are very similar, i.e., −0.113 eV Å$^{-1}$ for the initial structure with O–O distance to be 1.9 Å (blue) and −0.106 eV Å$^{-1}$ for O–O distance of 2.1 Å (red), indicating good convergence of our PMF calculations. To calculate the entropy change, we firstly fit the temperature dependent free energy curves, and then the entropy was obtained by taking the temperature derivative. Similarly, the heat capacity was obtained by taking the temperature derivative of the fitted canonical caloric curve $\langle E \rangle (T)$.

For comparison, the Gaussian 09 code[50] was also used for geometry optimization calculation of $O_2$ dissociation on $Au_{13}$. The PBE1PBE[51] functional was employed to optimize the geometries of Au-$O_2$ complex. The aug-cc-pvtz[52] and Lanl2DZ[53] basis sets were employed for O atom and Au atom, respectively. During the structure optimization, all the atoms (Au and O atoms) were allowed to relax. The vibrational frequency calculations were carried out to identify the stationary points and transition states (TS) with zero and one imaginary frequency, respectively. The intrinsic reaction coordinate (IRC) calculation was also performed to verify that the transition state connects correctly to the expected minima.

## Data availability

The data that support the findings of this study are available from the corresponding author upon reasonable request.

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

## Acknowledgements

We thank Y. Gao, H. You and H. Häkkinen for helpful discussion. Funding support was provided by National Natural Science Foundation of China (Grant Nos. 91745103, 2181101075, and 21621091).

## Author contributions

J.C. conceived and designed the project. J.-J.S. performed the calculation. J.C. and J.-J.S. analyzed the results and wrote the manuscript.

## Competing interests

The authors declare no competing interests.
