## [Peer Review File · Nature Communications]

REVIEWERS' COMMENTS:

Reviewer #1 (Remarks to the Author):

The authors reasonably addressed the concerns of the referees. I think it would be useful to expand the discussion (last 3 paragraphs or so) to make a specific point that this effect may be tough to observe if there exists an ensemble of particle sizes in a given sample as the smallest structures and the largest structures are much more rigid. Further, the nature of the support will drastically influence this as those containing strong interactions may force metals to adopt 2D structures that would minimize fluidity. In short, an honest discussion of where this effect may or may not be present and further the difficulties in experimentally identifying this should be added.

Reviewer #2 (Remarks to the Author):

I read the revised manuscript and comments to both reviewers. In general, I am content with the responses to the technical issues raised to both my questions and that of the other reviewer. On the other hand I still do not feel the authors have the claim to novelty that they think they do.

Review Comments (in black) and Responses (in blue)

Reviewer 1:

The authors present an AIMD study of O₂ activation on small Au clusters on MgO. Their major conclusion is that due to the solid liquid phase transition around room temperature there is a larger entropic component to the reaction free energetics. They conclude that for the first time they show a liquid phase like behavior of clusters leads to an important impact on the reaction free energetics. The calculations are competently executed and analyzed but unfortunately the authors have really not done their homework on the topic of choice.

Response: We thank the reviewer for agreeing that the level of our calculation is competent, but we disagree with the reviewer that we have omitted the progress in literature work. Note also that our proposed catalytic effect of solid-to-liquid phase transitions *during the course of elementary reaction steps* is new, and totally different from the liquid-like behavior of dynamic clusters in literature as mentioned by the reviewer. Our point-by-point responses to the comments are provided below.

Entropy and dynamics in gold clusters was first addressed by the Tsinghua/PNNL group.: Wang, Yang-Gang, et al. "Dynamic formation of single-atom catalytic active sites on ceria-supported gold nanoparticles." *Nature communications* 6 (2015): 6511.
Wang, Yang-Gang, et al. "The role of reducible oxide–metal cluster charge transfer in catalytic processes: New insights on the catalytic mechanism of CO oxidation on Au/TiO₂ from ab initio molecular dynamics." *Journal of the American Chemical Society* 135, no. 29 (2013): 10673-10683.

Related work on other metal systems with similar themes

Zandkarimi, Borna, and Anastassia N. Alexandrova. "Surface-supported cluster catalysis: Ensembles of metastable states run the show." *Wiley Interdisciplinary Reviews: Computational Molecular Science* (2019): e1420.

Zandkarimi, Borna, and Anastassia N. Alexandrova. "Dynamics of Subnanometer Pt Clusters Can Break the Scaling Relationships in Catalysis." *The journal of physical chemistry letters* 10, no. 3 (2019): 460-467.

Liu, Xingchen, Xiaodong Wen, and Roald Hoffmann. "Surface Activation of Transition Metal Nanoparticles for Heterogeneous Catalysis: What We Can Learn

from Molecular Dynamics." ACS Catalysis 8, no. 4 (2018): 3365-3375.

These papers (amongst MANY others for these groups and others) clearly also show the liquid like behavior of supported Au clusters and the impact of entropy and dynamics upon catalysis . This body of work though well cited is completely ignored by the authors despite the similar perspective, method etc. For this alone I believe the paper has a lot less novelty than the authors claim as they have either ignored or are unaware of the earlier work by many other groups even on very similar systems.

Response: We thank the reviewer for the comments and suggestions of literature work. First of all, we'd like to stress that we are aware of the computational studies on structural dynamics of clusters, as suggested by the reviewer. This should be evident in our original manuscript. For example, in Lines 29-32, Page 1 in the original manuscript, we have stated, "*In recent years the advent of in situ spectroscopic and microscopic techniques and **electronic structure calculation methods** has allowed for investigating the dynamic evolution of the structures of catalysts under reaction conditions⁹⁻¹³*" and the most relevant experimental work, i.e. Refs. 9-13, have been cited. In addition, in Lines 55-56, Page 2, we have also cited computational work investigating temperature effects on structural dynamics of clusters, including one early AIMD work (Krishnamurty et al. JPCA 2007) and one more recent work which studied metal clusters at realistic temperatures by global optimization approach (JCTC 2016) from the group of Alexandrova (one of the groups also mentioned by the reviewer). We regret that due to length limit, we could not cite all computational publications in this topical area even though we've tried to cite the most relevant ones. In respect for the reviewer's suggestions, we have accordingly cited these references in the revised manuscript (Refs. 14-15 and 17-19 on page 1).

Secondly, we respectfully disagree with the reviewer's statement "*the paper has a lot less novelty than the authors claim as they have either ignored or are unaware of the earlier work by many other groups even on very similar systems*". We emphasize that the novelty of our work doesn't lie at ab initio molecular dynamics (AIMD) simulations of structural dynamics of Au clusters, which is the focus of most of previous computational studies, but on the systematic investigation of temperature dependence of reaction free energies (barriers) on dynamical clusters, which enables us to correctly extract reaction entropies due to cluster dynamics, most importantly

leading to the discovery of *solid-to-liquid phase transitions of Au clusters during the course of elementary reactions* that can facilitate chemical transformations. We are very confident that this discovery is new, and has never been proposed before. Our claim is also supported by the second reviewer who, after summarizing three of our main findings, concluded that “*the general concept is very interesting...*” and our studies “*highlight the potential impact of the effect on predicted reaction and activation free energies*”.

More specifically, previous computational studies on cluster dynamics concern the diversity of structural configurations of given sizes of clusters, and AIMD (e.g. the work of Jun Li et al. as indicated by the reviewer) and global optimization techniques (e.g. the work of Alexandrova) have been applied to exploring the configuration spaces to search for (meta-)stable structures. Once stable structures are obtained, reaction barriers and energies are calculated on given cluster structures using ‘static’ geometry optimization methods (e.g. transition state optimization methods, etc.). Thus, as far as elementary reactions are concerned, the cluster structures remain ‘static’. In this sense, structural dynamics of clusters just brings in many stable structures and ‘static’ reaction paths on each one of them. For this reason, we have argued, as described in Lines 32-34, Page 1 in the original manuscript, “*the “static” perspective has been still taken in identifying the active sites under environmental conditions and monitoring the transformation of one type of active sites into another*”. This approach can be justified if the time scale of structural evolution of catalysts is much longer than the time scale of elementary reaction steps.

Our work however goes beyond this common view in literature, aiming to address a fundamental question, what if the timescale of the dynamic evolution of catalyst structures overlaps with that of chemical reactions? Thus, we take a distinctly different standpoint from previous publications, as also indicated by Review 2 stating that “*the authors correctly argue that most in the field do not consider active site dynamics on the time scales of reaction events*” and “*the methodology of huge numbers of AIMD simulations is pushing the boundaries of studying the dynamic behaviour of catalytic systems*”. In our work, we have carried out extensive AIMD simulations to calculate reaction free energy profiles by using constrained MD and thermodynamic integration, fully accounting for cluster dynamics during the course of

elementary reactions. Moreover, careful investigations of temperature dependence of the reaction free energies enable us to quantify the entropic contributions due to cluster dynamics, which eventually leads to the discovery of the new catalytic effect that solid-to-liquid phase transitions of the Au clusters induced by the change of adsorption states of reaction species facilitate chemical transformations.

To avoid any confusion, we have added some sentences to clarify the advance and novelty of our work citing previous computational studies as suggested.

On Page 1, we now write: *“Although in recent years the advent of in situ spectroscopic and microscopic techniques⁹⁻¹³ and electronic structure calculation methods has allowed for investigating the dynamic evolution of the structures of catalysts under reaction conditions¹⁴⁻¹⁹, the “static” perspective has been still taken in identifying the active sites under environmental conditions and monitoring the transformation of one type of active sites into another. For example, ab initio molecular dynamics (AIMD) simulations have been used to explore the diversity of structural configurations of clusters, and however elementary reactions have been still calculated on given cluster structures using static optimization techniques at 0 K¹⁵. This is justifiable ...”*

Other points

A, The authors also need to have a much clear discussion of their method results and conversions. They are claiming to get an entropy by numerical differentiation of temperature dependent PMF. This is hard to do even with classical mechanics where the trajectories can be sampled much better. I would like to see error bars on the PMF as well as the entropy curves in Figure 1d

Response: We thank for the reviewer for the comments and suggestions. In the work, we have calculated the reaction free energy profiles by combining AIMD, constrained MD and thermodynamic integration. These free energy calculation methods are well established, and often used in solution reactions¹⁻⁴ and biological processes⁵. The reaction free energies (barriers) can be obtained by integrating potentials of mean force (PMF) over a chosen reaction coordinate, i.e. the O-O distance of O₂ dissociation reaction. Following the reviewer’s suggestion, we have added more discussion on our calculation results and conversions (Method section, highlighted in yellow).

Regarding the statistical errors in our free energy calculations, we would like to first point out that we usually run several ps AIMD simulations to equilibrate the systems, followed by production periods of about 5-15 ps for data analysis. For the size of systems studied in this work, the time scale of ~ 10 ps is sufficient to obtain well converged PMFs, as evident by the time accumulating averages shown in Extended Data Figs. 3, 5, 7. To quantify the error bars as suggested by the reviewer, we have divided the trajectories into five evenly spaced blocks, and calculated the average PMFs and free energies in separate, and then used the standard deviations of these five blocks of data as measure of statistical uncertainties. Consistent with good convergence of time accumulating averages, the estimated statistical uncertainties are indeed negligible, on the order of 0.03 eV/\AA for forces and 0.01 eV for energies. Taking O_2 dissociation on Au_{13} at 330K for example, the error bars of PMF and free energy profile are shown in Figure R1a and b below. The statistical uncertainties of PMF and free energy profiles at other temperatures have also be evaluated, as shown in Figure R1c, d. We have accordingly modified the manuscript to show that the statistical errors in our calculations are small.

Figure R1. **a**, Averaged force-distance curve with error bars of O_2 dissociation on Au_{13} at 330 K. **b**, Free energy profile with error bars of O_2 dissociation on Au_{13} at 330

K. **c**, Calculated averaged force-distance curves with error bars under various temperatures. **d**, Free energy profiles with error bars of O₂ dissociation on Au₁₃ under different temperatures.

As for the numerical differentiation of temperature dependent reaction free energies, it is indeed not straightforward, as pointed out by the reviewer. It is worth mentioning: (i) to obtain converged PMF at one fixed O-O distance, it usually takes 10-20 ps, i.e. 20000-40000 AIMD steps; (ii) to obtain accurate a free energy profile and integrated reaction free energy (barrier), we have calculated 15 distances from reactant to product states, with denser sampling around the transition states (TS), as shown in Figure R1a,c. This amounts to about half million AIMD steps for one free energy profile; (iii) to further study the temperature effect, we have investigated 7 temperatures well covering the temperature range of the solid, solid-liquid coexistence and liquid phases. Thus, it overall takes about 4-5 million AIMD steps to obtain sufficient data to assess the temperature dependence of the reaction free energies (Figure R2a). The error bar in the entropies, obtained by differentiating free energies with respect to the temperature, is however not straightforward to estimate. Note that it is not a statistical error, but a fitting error. The fitting at low and high temperature ranges are certainly more accurate, because of slowly varying of free energies, than that at the transition temperature range where free energies change more dramatically (see Figure R2 below). The fitting in the transition range will benefit from calculating more free energy points. If simply taking the two points at 273 K and 373 K and assuming a linear fit, the free energy change (i.e. points and curves in red) is about 1.2 eV, giving an average entropy change of ~1200 J/mol/K. This is of similar magnitude to the peak in the curve fitting. Most importantly, the sharp entropy change with such a magnitude clearly indicates the occurrence of phase transitions, irrespective of the fitting procedure of temperature dependent free energies. To clarify this, we have added some discussion in the revised manuscript (Method section, highlighted in yellow).

Figure R2. a, Temperature dependence of reaction free energies ($\Delta_r G$, red) and free energy barriers (ΔG^\ddagger , blue) with error bar. Free energies are referenced to those of the reactant states and the solid curves are the corresponding fits. **b**, Temperature dependence of the reaction entropy changes ($\Delta_r S$) and activation entropies (ΔS^\ddagger) calculated by AIMD.

B, The PMF at high temperature also look at bit strange almost as if there is a discontinuity around 2 Å are the authors sure that there is not a problem with the O-O distance as a collective variable?

Response: We thank the reviewer for the comment. It should be noted that the relatively steep decrease of mean forces around 2 Å at high temperature is exactly one would expect from sudden entropy change due to the melting of Au cluster after O₂ dissociation. If looking at the trend in the change of PMF profiles when increasing the temperature (see Figure R1c above), the PMF profile around 2 Å gradually shifts to the left, and the peak height increases, then followed by steeper decrease. Thus, the PMF at high temperature fits well in this trend. Also, from the estimate of statistical uncertainty above, we are confident that these PMF profiles are well converged, and the statistical error can hardly affect the integrated free energies. Finally, since O₂ dissociation is under study, the O-O distance is the natural choice for the reaction coordinate.

C, The authors are using the blue moon ensemble method for estimating free energies they should site it approximately, Carter, E. A., Giovanni Ciccotti, James T. Hynes, and Raymond Kapral. "Constrained reaction coordinate dynamics for the simulation of rare events." *Chemical Physics Letters* 156, no. 5 (1989): 472-477.

Response: Thank the reviewer for mentioning the work. We are aware of this early work on blue moon sampling method. We have cited in the original manuscript (Ref. 34) the more recent development by Sprik and Ciccotti, J. Chem. Phys. 1998, 109, 7737, which is the one we use as implemented in the CP2K code. We have followed the reviewer's suggestion and added this reference in the revised manuscript (Ref. 50 in the revised manuscript).

D. Ref 18 is misleading because it refers to the Car-Parrinello approach to AIMD but the authors are actually doing Born-Oppenheimer MD not CPMD. They may consider the more appropriate by Payne, Mike C., Michael P. Teter, Douglas C. Allan, T. A. Arias, and JD Joannopoulos. "Iterative minimization techniques for ab initio total-energy calculations: molecular dynamics and conjugate gradients." *Reviews of modern physics* 64, no. 4 (1992): 1045. Or Marx, Dominik, and Jürg Hutter. *Ab initio molecular dynamics: basic theory and advanced methods*. Cambridge University Press, 2009.

Response: We thank the reviewer for the comment. We indeed use the Born-Oppenheimer MD throughout this work. Although the Car-Parrinello approach is different, it is common that the CPMD and related communities usually cite the seminal work of Car and Parrinello (PRL 1985) to honor their pioneering contribution that initiates the development of the field of AIMD, as also indicated by the name (i.e. CP2K) of the code we use. Due to the limitation of space, we have cited the second reference suggested by the reviewer in the revised manuscript (Ref. 25 in the revised manuscript).

In short, I would suggest the authors tone down the novelty of their claims, competently site the previous literature and submit it to a more specialized journal such as the journal of physical chemistry or ACS catalysis.

Response: We hope we have addressed the criticisms of the reviewer above, and now s/he is convinced that our work is distinctly different from the previous computational studies on cluster dynamics. Our discovery connecting solid-to-liquid phase transitions of clusters and chemical reactions is conceptually new, and will stimulate future experimental and theoretical studies and open up new opportunities in catalysis. In addition, we would like to point out that the proposed concept may have far

reaching impact on other fields such as enzyme catalysis. In chemistry, the general belief on chemical reactions is that they are ‘local’ and can be only affected by neighboring coordination environments since chemical bonds are short ranged in nature. In recent years, more and more evidence has emerged to show protein dynamics, which is of long range in nature, plays important role in enzyme catalysis. This poses intriguing questions to address: How global can the reactions get, and what are the fundamental mechanisms behind? Our work demonstrates that reactions on clusters can be globally affected by cluster dynamics through a solid-to-liquid phase transition mechanism, which in turn may shed light into enzyme catalysis and possibly beyond.

Reviewer 2:

This paper reports on AIMD studies of molecular bond activation (O_2 and H_2O) on small Au clusters as a function of temperature and a comparison of these results to bond activation studied by “static” geometry optimization. Using AIMD the contribution of the configurational entropy of the cluster along the reaction pathway could be assessed. The main findings are: (1) AIMD predicts temperature dependences in the reaction and activation free energies that are not seen in the static calculations, (2) there exist significant gradients in free energy of activation and reaction during the solid to liquid phase transition of the clusters, and (3) that the nature of the adsorbate (ie. O_2^* vs $2O^*$ vs $O—O^*$ TS) causes the clusters to melt at different temperatures, which causes significant changes in the activation and reaction entropy that induces the free energies gradients. These findings lead the authors to conclude that the cluster dynamics play a significant role in bond activation free energies, which cannot be captured by typically used “static” DFT calculations. The general concept is very interesting and the case studies presented highlight the potential impact of the effect on predicted reaction and activation free energies. The methodology of huge numbers of AIMD simulations is pushing the boundaries of studying the dynamic behaviour of catalytic systems. The authors correctly argue that most in the field do not consider active site dynamics on the time scales of reaction events. However, there are many details that are not well explained in the main text or SI and potentially significant limitations on the impact of these ideas for the operation of real catalysts. Below are comments and questions that start with the approach and then follow to the potential broad impact of the ideas.

Response: We thank the reviewer for the positive comments. Especially, the reviewer agrees that “most in the field do not consider active site dynamics on the time scales of reaction events”, and thus our work is distinct from literature that our AIMD simulations assess “the configuration entropy of the cluster along the reaction pathway”. More importantly, the reviewer thinks that the method of AIMD simulations is pushing the boundaries of studying the dynamic behavior of catalytic systems and the general concept we discover is very interesting. We also appreciate the constructive comments on technical details of our calculations and potential impact of the discovered concept on real catalytic processes, which would certainly

help improving the manuscript. Our point-by-point responses to the comments are given as follows.

1. The authors must do a better job of explaining how the AIMD simulations were done/justified and how the “static” geometry optimization calculations were done.

(a) In the case of the static calculations, was this done by identifying initial and final states, then running CI-NEB calculations to identify the TS and allowing all atoms in the cell to relax along the reaction coordinate? Or was the cluster kept static while the O-O bond was elongated? Certain atoms allowed to relax? This should be explained.

Response: We thank the reviewer for the comments and suggestions. The static geometry optimization calculations were performed using Gaussian 09 code. During the structure optimization, all the atoms (Au and O atoms) were allowed to relax. The vibrational frequency calculations were carried out to identify the stationary points and transition states (TS) with zero and one imaginary frequency, respectively. The intrinsic reaction coordinate (IRC) calculation was also performed to verify that the transition state connects correctly to the expected minima. We have accordingly added more detailed descriptions on our calculation methods in the revised manuscript (Method section, highlighted in yellow).

(b) The AIMD simulations seem also to be “static” just from a different reference point – the molecular bond distance. Here the molecule structure was fixed in space, while the cluster was allowed to move around the molecule into an optimum structure at each bond distance. Is this correct? Is there any evidence that this approach well represents the case where the whole system is allowed to evolve without any fixed coordinate species? The second approach would require much too long of a simulation to be reasonable, but there must be some argument that the approach used (fixed molecular bond distance) can represent a real bond breaking event.

Response: We thank the reviewer for the comments. We calculate the reaction free energies by performing constrained molecular dynamics to compute the potential of mean forces (PMF) for a specific reaction coordinate, followed by thermodynamic integration of the mean forces along this reaction coordinate. It is worth mentioning that the free energy calculation methods are well established and have been widely used to study solution chemistry¹⁻⁴ and biological processes⁵. In our work, the O-O

distance is the suitable choice for the O₂ dissociation reaction. PMF at a given distance is indeed calculated by constraining the O-O distance, but it should not be considered as ‘static’ since all other degrees of freedom of O and Au atoms are allowed to relax and sampled using constrained MD. Note that to get proper statistics it takes about 10-20 ps, i.e. 20000-40000 AIMD steps, to converge the mean force. For one PMF profile, we calculate overall 15 distances along the reaction coordinate, amounting to about half million AIMD steps to obtain one free energy profile.

(c) Extended Data Figure 3c needs more explanation. The exact nature of the integration of PMF to obtain G should be provided and justified.

Response: We thank the reviewer for the comments and suggestions. As mentioned above, the used methods have solid theoretical foundation, and are familiar to free energy calculation communities. The obtained PMF can be regarded as averaged thermodynamic forces acting on the reaction coordinate, and integrating the forces along the reaction path gives the reversible work driving the reaction from reactant to product state, i.e. the reaction free energy. The exact formulation of this theorem can be found in literature^{6,7} (e.g. Sprik, M. and Ciccotti, G., J. Chem. Phys., 1998, 109, 7737 and Chipot, C. and Pohorille, A., Springer Series in chemical physics, 2007). We have included some explanation on the theory and methodology in the revised manuscript (Method section, highlighted in yellow): “Briefly, by taking the derivative of the free energy $A(\xi)$ with respect to the reaction coordinate ξ one can obtain⁵² $\frac{dA}{d\xi} = \langle \frac{\delta H_{\xi}^F}{\delta \xi} \rangle$, where H_{ξ}^F is the Fixman Hamiltonian of the generalized coordinates. The derivative of the Hamiltonian with respect to ξ can be seen as an external force which is applied on the system to keep the reaction coordinate constant. In our calculations, the change of the Hamiltonian with respect to ξ is expressed as the Lagrange multiplier λ for constraining the distance⁵¹. Thus, to calculate the free energy of O₂ dissociation, we first performed constrained MD simulations to calculate the mean forces (i.e. Lagrange multipliers) for a set of O-O distances along the reaction coordinate from the reactant (e.g. O₂ molecule on Au₁₃) to product state (e.g. two separate O atoms on Au₁₃). In the constrained MD runs, we gradually increased the O-O distance with small increments, using the last structure of the previous MD

run as the initial configuration for the next run. The free energies were then obtained by integrating the average forces with respect to the distance.”

(d) In many cases in Extended Data Figure 3d the Force curves are quite noisy. How were these fits to obtain free energy profiles that are smooth?

Response: We thank the reviewer for the comment. It should be mentioned that our calculated PMF are well converged with very small statistical uncertainty, i.e. Figure R1 as given in the response to the comment of reviewer 1. The bumpy features in the PMF profile such as the one at 600 K (see Figure R3 below), simply indicate the fine structures in the corresponding free energy profile. Note also that the free energy change is obtained by integrating the mean force against distance, i.e. the integral area of the force-distance curve, resulting in a rather smooth free energy curve as shown in Figure R3b. In addition, we have calculated overall 15 points with small increments along the reaction coordinate of the O-O distance, well populating the free energy profile to achieve accurate free energies.

Figure R3. a, Calculated force-distance curve of O₂ dissociation on Au₁₃ at 600 K. **b,** Integrated free energy profile of O₂ dissociation on Au₁₃ at 600 K.

(e) Were any AIMD simulations repeated with significantly different initial conditions? If so, how comparable were the results?

Response: We thank the reviewer for the comment. Yes, we have tested the effect of different initial conditions on the mean force values, and our results have shown that the obtained values are very close, indicating good convergence of our PMF

calculations. When calculating the mean force at the O-O distance of 2 Å, we have started the structure models in both forward and backward directions, i.e. the models with O-O distances of 1.9 Å and 2.1 Å, respectively, as shown in Figure R4a below. The corresponding time accumulating averages of the mean forces are plotted in Figure R4b, and it is clear that the converged values are very similar, i.e. -0.113 eV/Å for the initial structure with O-O distance to be 1.9 Å (black) and -0.106 eV/Å for O-O distance of 2.1 Å (red). This indicates very small hysteresis in our thermodynamic integration. We have added some discussion in the revised manuscript to clarify this (Method section, highlighted in yellow).

Figure R4. a, Snapshots of the structures of Au₁₃-O-O with different O-O distances. i, O-O distance is 1.9 Å. ii, O-O distance is 2.1 Å. The balls in yellow and red represent Au and O atoms, respectively. **b**, Time accumulative averages of forces at the O-O distance of 2.0 Å with different initial structures.

(f) In general there needed to be a significantly expanded section in the supplemental materials associated with the methodology of the approach.

Response: We thank the reviewer for the helpful suggestions. We have therefore expanded the method section to describe more details on methods and computational setup in the revised manuscript. (Method section, highlighted in yellow)

2. Next the potential importance of these results in the context of catalytic processes is considered. It makes good sense in general that the fluxional nature of clusters must be considered to couple with reactions, but exactly how important this will be in real catalytic systems is questionable.

Response: We thank the reviewer for the helpful comments. We have to admit that this conceptually new catalytic effect of dynamic clusters presented in our work is on the basis of sole theoretical calculations, but using computational methods with solid foundation. The experimental realization or validation is however not yet possible because it has just been discovered theoretically. However, we want to argue that the importance of our work lies at exactly where the concern of the reviewer is, i.e., how significantly does this new theoretically proven concept impact real catalytic processes? Addressing this question will certainly stimulate future experimental studies. More importantly, our work also clearly shows the direction how experiment can realize or validate this effect, namely, by investigating the temperature effect on cluster dynamics and catalysis.

We have demonstrated in this work this new effect exists on Au clusters in gas phase and on an oxide support. The reviewer has actually suggested a number of other interesting factors, i.e. cluster sizes, support oxides, co-adsorption effect that could affect cluster dynamics and phase transition temperatures. As a matter of fact, we have already been studying some of these factors and found some interesting behaviors of clusters (see below). Therefore, we believe our work would open up a new avenue in catalysis with plenty of interesting and important scientific questions for both experimental and theoretical studies.

(a) The most important issue is that the structure of catalytic clusters is likely more confined than observed for the gas phase Au_{13} clusters here. Under reaction conditions, clusters will be coordinated to supports and will likely have non-negligible coverage of adsorbates. The authors partially address the support influence, although it is hard to directly compare Au_{13} and Au_8/MgO , as the cluster size is different. However, it is seen that the support confines the mobility of the cluster based on the much weaker influence of the nature of the adsorbate (O_2^* , O_2^{++*} , 2O^*) on the systems entropy. For supports that interact more strongly with metals (ie. TiO_2 , CeO_2 , FeO_x), one would imagine the support is even stronger at screening the influence of the adsorbate on the entropy changes along the reaction coordinate. Further, in a case where multiple adsorbates are on the same small cluster, which would not react simultaneously with any reasonable probability, the adsorbates would further act to “pin” the structure. Thus, while the case presented where the

entropic effect is maximized (Au_{13} in gas phase with 1 adsorbate) certainly highlights the importance of cluster dynamics on the reaction pathway, whether this effect exists under realistic catalytic conditions seems questionable.

Response: We are grateful for the constructive comments of the reviewer. Firstly, we absolutely agree that the support will have strong effect on the cluster dynamics and phase transition temperature. In this work, we compare the Au clusters in gas phase and on MgO support, and the results indeed suggest that MgO support constrains the dynamic effect of Au cluster, leading to smaller entropy changes in transition temperature range compared to the free standing cluster. However, it should be noted that the estimate of entropy change due to phase transition is still on the order of 500 J/mol/K, which is very significant for surface reactions. On the other hand, there is some evidence that shows the support effect is not just reducing the cluster dynamics and screening the adsorbate influence on the phase transition behaviour of the cluster along the reaction coordinate. On some active support such as TiO_2 , CeO_2 and FeO_x , structural fluctuation of clusters can even be boosted⁸⁻¹¹, especially when oxygen vacancies are present⁸. We therefore believe that the support has non-trivial effects on cluster dynamics, which will surely simulate future investigations.

With regard to the influence of adsorbate coverage, it is usually thought that the increase of coverage will actually enhance the dynamic effect^{9,10}. As shown in our present work, with the adsorption of one molecule, the Au cluster becomes more dynamical, as evident by the decrease in the melting temperature. It is therefore reasonable to argue that in practical catalytic reaction conditions the dynamic effect will become even more dramatic at high surface coverage.

[Redacted]

(b) From a similar perspective, there should be cluster size dependence for these effects. I imagine clusters below 3-4 atoms become more “rigid” and present less configurational entropy, and likely a similar phenomenon occurs at larger cluster sizes (>15 -20 atoms) where crystal structures begin to develop. Some comments on the size dependence would be useful, as any real catalyst will have a range of sizes and the reactivity would be of the collective behavior of these species.

Response: We thank the reviewer for the comments, and totally agree that the dynamic catalytic effect is size-dependent. Generally, small clusters below 7 atoms have less the isomers¹², resulting in small entropic contribution. Whereas, too large clusters have well defined crystal structures^{10,11}, and tend to lack structural flexibility necessary for adsorption induced change in cluster dynamics. Thus, clusters consisting of tens of atoms are more likely to show the dynamic effect. It is interesting to note that there is some experimental evidence that indeed shows that the best Au clusters for the CO oxidation consist of tens of Au atoms¹³. In this work, we have also found Au₁₃ cluster on which O₂ dissociation can be facilitated by a solid-to-liquid phase transition at certain temperature ranges. Our preliminary results also indicate that Au₁₉ cluster shows a similar dynamic effect (unpublished data), but to a smaller extent compared to Au₁₃. More systematic computational studies are required to unravel the size dependence of phase transition effect on reactions. Note that the size effect can also be investigated and validated by experiment. Following the reviewer's suggestion, we have added some discussion on size effect in the revised manuscript (Discussion section, highlighted in yellow).

(c) Finally, the authors have only 1 experimental data point supporting their analysis – the cluster melting temperature. Is there any evidence in literature of varying reaction or activation free energies as catalyst move from solid to liquid phase?

Response: We are grateful for the useful comment. Since this theoretically discovered catalytic effect due to phase transitions during the reactions is new, we don't think there are any experimental work that have claimed this effect. However, there is a possibility that this effect may have been observed in some experimental studies without being realized. In the following, we list some experimental observations that may suggest our discovered effect.

It is generally believed that the activity of Au cluster are highly size dependent, and only the Au clusters with a size less than 3 nm exhibit enhanced catalytic activity¹⁴. The fundamental reason however is not clear. Taking ultra-small (<3nm) supported Au clusters for example, Tsukuda and coworkers found the activity of Au clusters Au_n (n=10,18,25,39,89) for cyclohexane oxidation, increases with the increase of size, reaching the highest at n = 39, and thereafter decreases¹⁵. Other experiment also

showed that for clusters Au_n ($n < 7$), the activity is very low^{16,17}. The volcano-shaped size dependence cannot be explained solely by geometric factors, such as the density of under-coordinated Au atoms. Interestingly, it was found that the larger the cluster size, the higher the temperature required for the reaction¹⁴. Considering the cluster dynamics is also size and temperature dependent, the size dependent phase transition behavior of clusters may offer a possible explanation to these observations, i.e. volcano-shaped size dependence and increase of reaction temperature with cluster size.

In heterogeneous catalysis, many factors of catalysts such as cluster sizes^{14,18}, support oxides^{14,19}, and co-adsorption effects²⁰, have been proven to have significant impact on catalytic reactivity. Conventionally, they are often explained by geometric or electronic effects. As the reviewer has also suggested, these factors could also change the behavior of cluster dynamics^{8,10}, and thus the activity. Our proposed concept connecting phase transition of clusters and chemical reactions, therefore add a new dimension in understanding the factors relevant to catalytic activity.

Finally, we would like to stress again that our work also shows how the proposed effect could be realized in experiment, i.e. looking into the interplay between factors that affect cluster dynamics (e.g. cluster size, support, coverage and co-adsorption) and temperature. We have added some discussion in the revised manuscript to discuss this (Discussion section, highlighted in yellow).

3. Lastly there were a few recent papers on AIMD for the dynamic behaviour of clusters coupling to reaction pathways and supported liquid metal catalysts that I was surprised weren't cited or discussed. See Nature Chemistry 9, 862–867 (2017) and Nature Communications 6, 6511 (2015).

Response: We thank the reviewer for the suggestion. We have added these citations and some discussion accordingly in the revised manuscript (Refs. 14 and 16 in the revised manuscript).

Reference:

1. Ciccotti, G., Ferrario, M., Hynes, J. T. & Kapral, R. Constrained Molecular Dynamics and the Mean Potential for an Ion Pair in a Polar Solvent. *Chem. Phys.* **129**, 241–251 (1989).
2. Ciccotti, G., Ferrario, M., Hynes, J. T. & Kapral, R. Dynamics of ion pair interconversion in a polar solvent Dynamics of ion pair interconversion in a polar solvent. *J. Chem. Phys.* **93**, 7137 (1990).
3. Keirstead, W. P. & Wilson, K. R. Molecular dynamics of a model SN1 reaction in water. *J. Chem. Phys.* **95**, 5256 (1991).
4. Curioni, A. et al. Density Functional Theory-Based Molecular Dynamics Simulation of Acid-Catalyzed Chemical Reactions in Liquid Trioxane. *J. Am. Chem. Soc.* **119**, 7218–7229 (1997).
5. Boczeko, E. M., Brooks, C. L. First-Principles Calculation of the Folding Free Energy of a Three-Helix Bundle Protein. *Science* **269**, 393–396 (1995).
6. Sprik, M. & Ciccotti, G. Free energy from constrained molecular dynamics. *J. Chem. Phys.* **109**, 7737–7744 (1998).
7. Chipot, C. & Pohorille, A. Editors Free Energy Calculations. *Springer Ser. Chem. Phys.* (2007).
8. Wang, Y. G. et al. The Role of Reducible Oxide–Metal Cluster Charge Transfer in Catalytic Processes : New Insights on the Catalytic Mechanism of CO Oxidation on Au/TiO₂ from ab Initio Molecular Dynamics. *J. Am. Chem. Soc.* **135**, 10673–10683 (2013).
9. Wang, Y. G., Mei, D., Glezakou, V. A., Li, J. & Rousseau, R. Dynamic formation of single-atom catalytic active sites on ceria-supported gold nanoparticles. *Nat. Commun.* **6**, 6511 (2015).

10. Yang, H. et al. Size-dependent dynamic structures of supported gold nanoparticles in CO oxidation reaction condition. *Proc. Natl. Acad. Sci.* **115**, 7700–7705 (2018).
11. Wei, X. J. et al. Geometrical Structure of the Gold–Iron(III) Oxide Interfacial Perimeter for CO Oxidation. *Angew. Chem. Int. Ed.* **57**, 11289–11293 (2018).
12. Arslan, H. & Güven, M. H. Melting dynamics and isomer distributions of small metal clusters. *New J. Phys.* **7**, 60 (2005).
13. Herzing, A. A., Kiely, C. J., Carley, A. F., Landon, P. & Hutchings, G. J. Identification of active gold nanoclusters on iron oxide supports for CO oxidation. *Science* **321**, 1331–1335 (2008).
14. Haruta, M. et al. Low-temperature oxidation of CO over gold supported on TiO₂, α -Fe₂O₃, and Co₃O₄. *J. Catal.* **144**, 175–192 (1993).
15. Liu, Y., Tsunoyama, H., Akita, T., Xie, S. & Tsukuda, T. Aerobic Oxidation of Cyclohexane Catalyzed by Size-Controlled Au. *ACS Catal.* **1**, 2–6 (2011).
16. Sanchez, A. et al. When Gold Is Not Noble : Nanoscale Gold Catalysts. *J. Phys. Chem. A* **103**, 9573–9578 (1999).
17. Lee, S., Fan, C., Wu, T. & Anderson, S. L. CO Oxidation on Au_n/TiO₂ Catalysts Produced by Size-Selected Cluster Deposition. *J. Am. Chem. Soc.* **126**, 5682–5683 (2004).
18. Nørskov, J. K. et al. On the origin of the catalytic activity of gold nanoparticles for low-temperature CO oxidation. *J. Catal.* **223**, 232–235 (2004).
19. Casaletto, M. P., Longo, A., Venezia, A. M., Martorana, A. & Prestianni, A. Metal-support and preparation influence on the structural and electronic properties of gold catalysts. *Appl. Catal. A Gen.* **302**, 309–316 (2006).
20. Green, I. X., Tang, W., Neurock, M. & Yates, J. T. Spectroscopic Observation of Dual Catalytic Sites During Oxidation of CO on a Au/TiO₂ Catalyst. *Science* **333**, 736–739 (2011).